# Prevalence and correlates of prescription drug diversion and misuse among people living with HIV in the eThekwini district, KwaZulu-Natal, South Africa

**Buyisile Chibi**[1,2]*, **Nonhlanhla Yende-Zuma**[2], **Tivani P. Mashamba-Thompson**[1,3]

**1** Discipline of Public Health Medicine, School of Nursing and Public Health, University of KwaZulu-Natal, Durban, South Africa, **2** Centre for the AIDS Programme of Research in South Africa, Doris Duke Medical Research Institute, Durban, South Africa, **3** Department of Public Health, Faculty of Health Science, University of Limpopo, Limpopo Province, South Africa

* chibuyi@gmail.com

**Data Availability Statement:** All relevant data are within the paper and supporting information files.

## Abstract

### Background

Prescription drug diversion, and misuse has increased over the past decade and is notably in high-income-countries and significantly contributes to the opioid epidemic. People living with HIV (PLWH) are particularly vulnerable to prescription drug diversion, and misuse as most experience chronic pain, mental health problems and HIV-related illnesses. The researchers investigated the prevalence and correlates of prescription drug diversion, and misuse among PLWH in the eThekwini district, KwaZulu-Natal.

### Methods

A cross-sectional study was conducted among 392 PLWH, conveniently recruited from the public healthcare facilities located in rural, semi-urban and urban areas of the eThekwini district. Participants answered questions about their background, prescription medications, substance use, and prescription drug diversion, and misuse. Descriptive analysis was performed to estimate the prevalence of prescription drug diversion, and misuse. Multivariable logistic regression was used to identify predictors of prescription drug diversion, and misuse.

### Findings

Overall, 13% of the participants reported lifetime prescription drug diversion. The most common type of diversion was using prescription medication not prescribed by a healthcare provider (11%), followed by sharing of prescription medication (9%) and buying prescription medication without a medical script (5%). Twenty-three per cent of the participants reported prescription drug misuse in the past 90 days, with using prescription medication without a healthcare providers' guidance (9%) and not following the scheduled time periods (8%) being the most common reported types of misuse. Self-medicating was identified as a risk

**Funding:** Buyisile Chibi Grant no. 107620 National Research Foundation Buyisile Chibi Grant no. N/A University of KwaZulu-Natal, College of Health Sciences. The funders had no role in study design, data collection and analysis, decision to publish, or preparation of the manuscript.

**Competing interests:** The authors have declared that no competing interests exist.

factor for prescription drug misuse. There was no association between ART adherence and prescription drug diversion, and misuse.

## Conclusion

The study findings contribute to improving the limited data available on prescription drug diversion, and misuse among PLWH in South Africa. The prevalence underscores a need for urgent interventions when prescribing medications with potential risks. Addressing the risk of self-medicating is imperative for HIV care outcomes and to avert death.

## Introduction

### For the purpose of this study, the following terms were defined as

**Prescription drugs.**   Controlled medicines that would legally require a medical prescription from a healthcare provider in order to be dispensed due to their potential risks of being diverted, misused, abused and addiction.

**Drug.**   Refers to any substance which has a physiological effect or the potential to prevent or cure a disease. In this study it will be used interchangeably with medicines or medications.

**Drug diversion.**   Acquiring or distributing any legally prescribed controlled medicinal drugs to/from another person for any illicit use. Literature also reports drug diversion or medication diversion interchangeably alongside misuse, abuse, recreational use, non-medical use, illicit use, selling, sharing, giving-away, trading, stealing, missing/losing, doctor shopping and pharmacy shopping [1].

**Misuse.**   The inappropriate use of medicine or not adhering to the information leaflet of the medicine. This includes the following: without a health care provider's guidance or instructions, non-medically (e.g. used it recreationally), without following the dosage instructions, not following the correct way of administration of the medication, not following the scheduled time periods to further reduce the amount of pain you were feeling, to increase the ability to concentrate, to boost feelings and performance enhancing, to experience pleasure or excitement and intense feelings of well-being and happiness and to get high [2]. However, according to the National Institute on Drug Abuse, medication misuse is defined as use of medication in ways or amounts other than intended by a doctor, by someone other than for whom the medication is prescribed, or for the experience or feeling the medication causes. This term is used interchangeably with "non-medical" use, a term employed by many national drug use surveys [3].

**Self-medication.**   The use of medicinal drugs without the guidance of a health care provider [3].

Global efforts are being made to end the AIDS epidemic, but countries have reported mixed results [4]. In 2018 about 37.9 million people were living with HIV worldwide, with 1.7 million new HIV infections and 770 000 deaths [5]. Approximately 23.3 million people living with HIV (PLWH) were receiving antiretroviral therapy, but only 53% were virally suppressed [5]. Since the universal test and treat, the 90-90-90 target, as well as the UNIADS fast track strategy, only about 57% AIDS event and death reduction has been achieved with early ART initiation [6]. South Africa (SA) accounts for 19% of the global HIV prevalence rate, [5] and the KwaZulu-Natal province has the highest HIV prevalence within the country [7]. Barriers to achieving and maintaining viral suppression have been documented, one of which is prescription drug diversion, and misuse [1,8]. One of the dangers of prescription drug diversion

is transitioning into illicit drug use, worsening the barriers of HIV care and challenging the HIV/AIDS targets to put a stop to the HIV epidemic. A recent scoping review study revealed that prescription drug diversion among PLWH came into existence as early as 1995; however, there is a lack of evidence in resource-limited-settings with high HIV rates [9].

PLWH often experience illnesses that correlate with prescription drug diversion, and misuse. Evidence suggests that chronic pain is the common comorbidity among PLWH that correlates with prescription drug misuse [10,11]. Literature has also shown substance use disorders or dependence as risk factors for prescription drug diversion among PLWH [8,10,12–21]. Prescription drug diversion, and misuse have been associated with mental health problems such as depression symptoms, psychiatric disorder and anxiety as risk factors among PLWH [16,18,20,22–24]. Other factors associated with prescription drug diversion, and misuse among PLWH included being male [16,20], young in age [14,22], HIV-related stigma [16], poor in health [18], lower in the level of education [19,24], unemployed or having no income [25], having an experience of diverting, and misusing medicines [15,26], owning addictive prescriptions [10,24,26], and living in high-level neighbourhood disorder [19]. In the United States, these factors were reported as being associated with new HIV infections [27]. A recent scoping review study showed that there is limited research on prescription drug diversion in low-income-countries among PLWH [28].

Research investigating medication misuse among the general population can reveal and guide how prescription drug diversion, and misuse might unfold among PLWH in our local context. Our local context is surrounded by several challenges related to consumer, health care providers and drug treatment centres. Challenges related to the user includes: codeine misusers did not know of their dependency and regarded their use as appropriate [29]. Codeine misusers were also unaware of the addictive potential of codeine, including the risk of dependence and harm [29]. Another challenge related to the user was the ability to read and understand the medicine information leaflet [30]. About 29% of people accessing medicines from the SA retail pharmacies reported not reading the medicine information leaflet [30]. Most people perceived codeine-containing products as safe medications, while only a few people acknowledged the harmfulness and addictiveness of these medicines [30]. Health care providers also have their share of challenges. A study conducted in SA showed that about 95% of health care providers prescribing codeine products raised a concern that patients do not understand the risks of medicine dependence since they perceived it as safe [31].

Furthermore, health care providers perceive that patients were inadequately given information on medicine usage [31]. Another challenge faced by health care providers was the difficulty to identify problematic medicine use without the patient disclosing of their addiction problem [31]. A study by Carney *et al.* (2018), reported that pharmacy staff were challenged by lack of specialist training in substance use issues, therefore, making it challenging to curb medicine misuse [32]. Dada *et al.* (2015) study among the South African Community Epidemiology Network on Drug Use (SACENDU) identified a gap that codeine might be used as a substitute for heroin in SA when the heroin is unavailable, therefore worsening the problem of drug diversion [33]. On the other hand, drug treatment centres in SA are mostly located in urban areas which lack tracking of codeine misuse and dependence [34]. Additionally, a qualitative study in SA revealed that ARVs were crushed and added to illicit drugs and smoked for their psychoactive effects [35].

Data on the extent of prescription drug diversion, and misuse among PLWH are very limited, despite the emerging public health problems and the high HIV prevalence in the KwaZulu-Natal province. This study aims to investigate the prevalence and correlates of prescription drug diversion, and misuse and among PLWH in the eThekwini district, KwaZulu-Natal, as well as the association between ART adherence and prescription drug diversion,

and misuse. Knowing the extent and correlates associated with prescription drug diversion and misuse could inform healthcare providers and policymakers to incorporate appropriate and relevant interventions to address problems when managing HIV.

## Methods

### Study design

This study comprised of a cross-sectional survey conducted in the eThekwini district, Kwa-Zulu-Natal province, South Africa, and forms part of a sequential, explanatory mixed-method approach involving a cross-sectional survey and in-depth interviews. The researchers surveyed to investigate the extent of prescription drug diversion, and misuse among PLWH in the eThe-kwini district from August 2018 to May 2019. An interviewer-administered questionnaire was used to collect data from a convenient sample of PLWH who, at the time of the survey, were at the recruitment site.

### Study setting

South Africa is situated on the African continent and made up of nine provinces. The entire study was conducted in the eThekwini district, KwaZulu-Natal province, located in the south-east on the country. eThekwini is one of eleven districts in KwaZulu-Natal province. According to Statistics SA, the population size was estimated at 3.4 million in 2011 [28]. The majority of its people speak isiZulu (62%), and 26% are English language speaking. About 73% of the residents are Black African, 16% Indian/Asian, 6% White and 2.5% Coloured. The Fifth South African National HIV Prevalence, Incidence, Behaviour and Communication Survey (SABSSM V) reported that the HIV prevalence in KwaZulu-Natal was the highest of all provinces at 27.0% (95% CI 23.9–30.4) in 2017. Two of the key challenges to eThekwini district's health care service provision is the high prevalence of HIV/AIDS and also the abuse of drugs and alcohol [36]. Until now, the extent of prescription drug diversion, and misuse in the eThe-kwini district is, however unknown.

### Sampling method

The researchers employed a stratified sampling method for sampling public healthcare clinics as recruitment sites for PLWH. The researchers grouped all eThekwini district public health-care clinics providing HIV care programme into strata based on geographic settings, for example, rural, urban, and semi-urban. Overall, the eThekwini district has 118 public healthcare facilities, of which 6% are in rural areas, 53% in urban areas and 41% in semi-urban areas. Stratified randomisation of public healthcare clinics in the three strata was ensured that the proportions of the sample matched those from all three geographical settings. Permission to conduct the survey at each clinic was granted through relevant authorities and clinics that were unable to give permission to participate in the study were excluded. The researchers con-veniently approached and invited people who were waiting for healthcare worker consulta-tions at the clinics to participate in the study. To reach people who were drug users and living with HIV, we employed an outreach and recruitment approach through a Non-Governmental Organisation (NGO). The NGO had an existing and on-going outreach programme which supported drug users around eThekwini. They had programmes in place such as HIV testing, linkage to care and harm reduction interventions (including the provision of clean syringes and needles and opioid substitution therapy), and permission was obtained to join their out-reach team in order to recruit drug users who might be interested in participating in our study.

## Participants

Three hundred and ninety-two people who verbally confirmed to be living with HIV and were aged 18 years or older, gave written consent. Some participants were assisted in their preferred language during consenting and conducting the survey. All participants were in the eThekwini district at the time of the survey.

## Data collection and instruments

The researchers adapted the cross-sectional survey tool from a study conducted by Tsuyuki and Surratt (2015) [37] to fit our population, setting and study aim. We used hard copies of the survey which consisted of ten categories: background, prescription medications (PM), substance use, prescription drug diversion, non-diversion, prescription drug misuse, impressions of PM black market, antiretroviral therapy adherence, and stress, mental and emotional health and stigma. Data were collected by the researcher and a trained research assistant. A pilot test study, to ensure the validity of the tool among PLWH, was conducted before administering the questionnaire and helped us to improve precision, reliability, validity of data, identify problems/omissions, and assess time spent when completing the survey. The survey was conducted in English; however, questions were clarified to the participants in their preferred language. The survey lasted 60 minutes or less.

## Ethical considerations

Full ethical clearance from the University of KwaZulu-Natal's Biomedical Research Ethics Committee (Reference No. BE666/17), the KwaZulu-Natal Provincial Department of Health (Reference No. HRKM055/18, KZ_201802_026) and recruitment site clearances were obtained before commencing our data collection. Questionnaires were identified by study ID (identity) numbers and not linked to consent forms. The dynamics of each clinic was obtained from the facility manager and the nurses responsible for HIV management before conducting the survey to minimise harm and stigma when inviting potential participants to participate in the study. A private space was allocated to maintain confidentiality during data collection, data collection date was set, a tour around the facility and introduction to other healthcare providers were all done in preparation for the data collection. On the day, the facility manager or the nurse in charge made a brief introduction to the patients in the waiting area before data collection commenced. During the consenting process, researchers explained extensively to each participant that their participation was voluntary, that the study was not associated with the clinic, and participation would not affect their healthcare. Participants were assured that their responses would be identified with study ID numbers, not their names.

## Measures

**Outcomes.**    This study focused on investigating the prevalence of and factors influencing prescription drug diversion, and misuse among PLWH. The researchers used the Gelberg–Andersen Behavioural Model to guide the development of the survey questions and operationalization in this study. This conceptual framework has been fully studied for healthcare utilisation for vulnerable populations [38].

**Our analysis examined three outcome measures:** The first outcome was prescription drug diversion which stemmed from the question: "Have you ever diverted (sold, borrowed, shared, gave-away, visited multiple doctors/pharmacies for the same medications or traded or stole) prescription medication (yes/no)?" Prescription drug diversion was then recoded to "have you ever diverted prescription drugs in your lifetime (yes/no)?".

The second outcome was prescription drug misuse which stemmed from several questions investigating different types of drug misuse: "In the past 90 days, have you ever used prescription medication (PM): without a healthcare provider guidance or instructions, not following the scheduled time periods, non-medically, to further reduce pain, without following the dosage instructions, not following the correct way of administration, to enhance performance, for pleasure or to get high?" Prescription drug misuse was recoded to "in the past 90 days, have you ever misused prescription drugs (yes/no)?".

Additionally, our questionnaire incorporated the AIDS Clinical Trials Group (ACTG) adherence questions to measure per cent ART adherence. Per cent ART adherence was measured using the self-reported data on ART adherence in the past seven days, which stemmed from:

- How many pills (ARVs) are you prescribed to take each time?

  ◦ Number of pills per day

  ◦ Number of pills per time

- How many doses did you miss in the past seven days?

- The per cent ART adherence was calculated by the total weekly ARV doses taken divided by the total ARV doses that were prescribed, converted to a percentage

## Explanatory variables

- Demographics assessed whether age, gender, marital status, level of education, employment, self-rated health, neighbourhood one resides at and homelessness had an influence on prescription drug diversion or misuse. Disorderly neighbourhoods and homelessness were hypothesized to add vulnerability to individuals which then influences prescription drug diversion or misuse.

- Self-medicating for conditions that one was diagnosed with (coded as yes/no) assessed whether self-medicating had any influence or facilitated prescription drug diversion or misuse.

- Owing prescription drugs or being in possession of prescription drugs (coded as yes/no) assessed whether owing or being in possession with prescription drugs had any influence or facilitated prescription drug diversion or misuse.

- Being educated or trained on how to use or take any prescription medication (coded as yes/no) assessed whether education or training acted as a protective factor from prescription drug diversion or misuse.

- Being knowledgeable that prescription drugs are harmful to one's health if not used as prescribed assessed whether knowing the risk of prescription drugs would protect a person from diverting or misusing the medication.

- Substance use (included alcohol, marijuana, tobacco and illegal drugs) which stemmed from a question "when was the last time that you used alcohol/marijuana/tobacco/illegal drugs?" This was then recoded to, "have you ever used alcohol/marijuana/tobacco/illegal drugs in your lifetime (yes/no)?". Substance used is hypothesized to add vulnerability to individuals which then influenced prescription drug diversion or misuse.

## Data management and analysis

Data were captured into a password protected excel spreadsheet with study ID numbers. The accuracy of source data was ensured through double capturing, with the researcher as the

second data capturer. After data cleaning and verification, data were imported into The R Project for Statistical Computing R version 3.5.1 (2018-07-02) [39] for analysis. All source documents were stored in a secured place as back up.

Outcome variables were coded as follows: 1 = ever diverted prescription drugs vs 0 = never diverted prescription drugs; 1 = misused prescription drugs in the last 90 days vs 0 = never misused prescription drugs in the past 90 days.

Continuous variables were summarised using means with standard deviation and compared across the two outcomes variables (diversion and misuse), using either independent samples t-test or Wilcoxon rank-sum test. Categorical variables were reported as frequencies with percentages and compared using Fisher's exact test. The prevalence with exact 95% confidence interval was calculated for prescription drug diversion, and misuse among PLWH in the eThekwini district.

Two types of multivariable logistic regression models were performed to identify variables that were associated with diversion and misuse. The association of each variable on the outcome (diversion or misuse) controlling for the geographical setting was assessed for model A. Model A represent the odds ratios showing the association of each variable on diversion or misuse among PLWH in the eThekwini district and detailed information presented in the S2 Table. All variables were fitted in the model, including geographical setting, to assess their collective association on the outcome for model B. Model B represents adjusted odds ratios showing the association of a group of variables on diversion or misuse among PLWH in the eThekwini district and detailed information presented in the S3 Table. Since the study had too many variables to use in model B, age and gender were selected as these variables are known to be confounders of many exposure-outcome relationships [40,41]. All variables that were plausible enough to be associated with each of the outcome variables were then included in model B. We regarded this strategy as well thought out and scientifically rigorous, as opposed to using p-values to reduce the number of variables.

The following variables were included in model B for prescription drug diversion outcome: age, gender, education, income, substance use (tobacco and illegal drugs), and self-medicating. The variables included in model B for prescription drug misuse were age, gender, homelessness, substance use (tobacco and illegal drugs), self-medicating and awareness of the black market.

The percentage of ARV adherence was calculated by the total weekly ARV doses taken divided by the total ARV doses prescribed, converted to a percentage. The analysis was conducted to assess the association between the percentage of ARV adherence with prescription drug diversion or misuse.

All analyses were performed using R studio, and Microsoft Excel was used to plot graphs and forest plots.

**Justification for not performing ad hoc corrections to correct for multiple comparisons.** Our analyses were exploratory, and there was no a priori hypothesis being tested. We were of the view that either committing type 1 error (false positive) or type 2 error (false negative) in our analyses is not critical. Our reasoning was also based on the premise that we are not testing any treatment nor performing any analyses that can potentially change guidelines, so that is why we did not correct for multiple comparisons.

## Results

### Characteristics of study participants

The study received a 100% response rate since all persons approached to participate agreed and completed the survey. Completed responses were received from 392 HIV positive

participants, recruited from three geographic settings: rural setting 6% (n = 25), urban setting 58% (n = 226) and semi-urban setting 36% (n = 141). The demographics of the participants were females 67% (n = 263), males 33% (n = 129), average age 36.78 (± 10.0) years, African 99% (n = 388), single 86% (n = 335) and reported being homeless in the past 30 days 4% (n = 17) (Tables 1 and S1 in the supplementary material file). Eighty-two per cent of participants were on antiretroviral treatment, with an average of 96% of ART adherence in the past seven days, 61% (n = 238) have not completed high school, and 53% (n = 209) were employed.

## Rates of prescription drug diversion and its predictors

Overall, 13% (n = 51) of the sample reported ever diverting prescription drugs (Fig 1). About 11% (n = 42), 9% (n = 36), and 5% (n = 20) of the participants reported using PMs not prescribed by a healthcare provider, sharing PMs, and to have bought PM from someone else or from a pharmacy without a medical script for their own use in the past 90 days, respectively. The least type of diversions reported, at 2% (n = 9) or less, were giving away, borrowing, and trading of PM.

Correlations between characteristics of participants who have diverted versus those who have never diverted prescription drugs are shown in Table 1. In the multivariable model A (Fig 2) gender, income and tobacco use were associated with prescription drug diversion. Males had 2-fold increased odds of diverting PM compared with females [OR = 2.01 (95% CI: 1.10–3.65), $p$ = 0.022]. Participants who had not received an income in the past month had 2-fold increased odds of diverting when compared to those who have received income in the past month [OR = 2.02 (95% CI: 1.08–3.80), $p$ = 0.029]. Participants who used tobacco had an almost 3-fold higher chance of diverting PM compared to non-substance users [OR = 2.74 (95% CI: 1.50–5.02), $p$ = 0.001]. Participants who used marijuana and illegal drugs had 2-fold higher chance for diverting PM, [OR = 2.05 (95% CI: 0.99–4.23), $p$ = 0.052] and [OR = 2.69 (95% CI: 0.98–7.38), $p$ = 0.054].

In the multivariable model B, there was no association in participants with no income, have used tobacco, used illegal drugs, and have self-medicated (S2 Table).

## Rates of prescription drug misuse and its correlates

Overall, 23% (n = 89) of the sample reported having misused prescription drugs in the past 90 days (Fig 3). Among all the participants in the study, 9% (n = 35) and 8% (n = 33), reported to have used PM without a healthcare provider's guidance and not following the scheduled time periods in the past 90 days, respectively. In addition, among the participants who misused drugs, 6% (n = 25), 5% (n = 18), 4% (n = 14), reported to have used prescription drugs non-medically, to further reduce the amount of pain, and without following the dosage instructions in the past 90 days. The least reported types of misuse, each reported at 1% (n = 5), were not following the correct ways of administering the medication to increase the ability to concentrate and boost feelings; and performance-enhancing to experience pleasure and to get high.

The characteristics of participants who have misused versus those who have never misused prescription drugs were compared (Table 1). Participants who used illegal drugs had more than 3-fold higher odds of misusing PM [OR = 3.37 (95% CI: 1.37–8.30), $p$ = 0.008] when compared to those who do not use illegal drugs. Participants who were self-medicating for diagnosed conditions had almost 3-fold higher odds [OR = 2.91 (95% CI: 1.62–5.21), $p$ <0.001] of misusing PM (Fig 4).

Results from model B showed that self-medicating increased the odds of misusing PM [OR = 2.63 (95% CI: 1.44–4.82), $p$ = 0.002]. Participants with the following characteristics:

**Table 1. Participants characteristics stratified by prescription drug diversion and misuse.**

| Characteristics | Total N = 392 | Ever diverted prescription drugs, N = 51 | Did not divert prescription drugs, N = 341 | p value | Ever misused prescription drugs, N = 89 | Did not misuse prescription drugs, N = 303 | p value |
|---|---|---|---|---|---|---|---|
| Age in years, mean ±SD | 36.8 ± 10.0 | 37.1 ± 8.5 | 36.7 ± 10.2 | 0.807 | 35.4 ± 8.6 | 37.2 ± 10.4 | 0.103 |
| **Age groups, n (%)** | | | | | | | |
| 18–24 years | 27 (7) | 4 (8) | 23 (7) | 0.2487 | 7 (8) | 20 (7) | 0.8149 |
| 25–29 years | 79 (20) | 6 (12) | 73 (21) | | 19 (21) | 60 (20) | |
| 30 + years | 286 (73) | 41 (80) | 245 (72) | | 63 (71) | 223 (73) | |
| **Gender, n (%)** | | | | | | | |
| Male | 129 (33) | 24 (47) | 105 (31) | **0.032** | 29 (33) | 100 (33) | 1.000 |
| Female | 263 (67) | 27 (53) | 236 (69) | | 60 (67) | 203 (67) | |
| **Education groups, n (%)** | | | | | | | |
| Not completed high school | 238 (61) | 34 (67) | 204 (60) | 0.6305 | 55 (62) | 183 (60) | 0.694 |
| Completed only high school | 114 (29) | 13 (25) | 101 (30) | | 25 (28) | 89 (29) | |
| Completed both high school and tertiary studies | 40 (10) | 4 (8) | 36 (10) | | 9 (10) | 31 (10) | |
| **Employment status, n (%)** | | | | | | | |
| Employed | 209 (53) | 25 (49) | 184 (54) | 0.549 | 44 (49) | 139 (46) | 0.629 |
| Unemployed | 183 (47) | 26 (51) | 157 (46) | | 45 (51) | 164 (54) | |
| **Participants who received income past month, n (%) [a]** | | | | | | | |
| No | 90 (23) | 18 (35) | 72 (21) | 0.041 | 20 (22) | 70 (23) | 0.991 |
| Yes | 300 (77) | 33 (65) | 267 (78) | | 69 (78) | 231 (76) | |
| **Homeless, n (%)** | | | | | | | |
| No | 374 (95) | 48 (94) | 326 (96) | 0.716 | 82 (92) | 292 (96) | 0.144 |
| Yes | 18 (5) | 3 (6) | 15 (4) | | 7 (8) | 11 (4) | |
| **Substance use, n (%)** | | | | | | | |
| **Ever used alcohol [b]** | | | | | | | |
| No | 205 (52) | 23 (45) | 182 (53) | **0.410** | 49 (55) | 156 (51) | 0.567 |
| Yes | 186 (48) | 27 (53) | 159 (47) | | 39 (44) | 147 (49) | |
| **Ever used tobacco [c]** | | | | | | | |
| No | 285 (73) | 27 (53) | 258 (76) | **0.001** | 61 (68) | 224 (74) | 0.360 |
| Yes | 106 (27) | 24 (47) | 83 (24) | | 28 (32) | 78 (26) | |
| **Ever used marijuana [d]** | | | | | | | |
| No | 335 (86) | 39 (76) | 296 (87) | 0.063 | 72 (81) | 263 (87) | 0.171 |
| Yes | 55 (14) | 12 (24) | 43 (13) | | 17 (19) | 38 (13) | |
| **Ever used illegal drugs** | | | | | | | |
| No | 371 (95) | 45 (88) | 326 | **0.042** | 79 (89) | 292 (96) | **0.012** |
| Yes | 21 (5) | 6 (12) | 15 (4) | | 10 (11) | 11 (4) | |
| **Perceived misusing or abusing of prescription medication not harmful their health, n (%) [e]** | 66 (17) | 8 (16) | 58 (17) | 0.495 | 13 (15) | 53 (18) | 0.725 |
| **% ART adherence, mean ±SD [f]** | 96.4 ± 12.9 | 94.5 ± 16.0 | 96.7 ± 12. | 0.366 | 96.0 ± 11 | 96.5 ± 13.3 | 0.705 |
| **Diagnosed with other conditions, n (%)** | | | | | | | |
| No | 274 (70) | 36 (71) | 238 (70) | 1.000 | 64 (72) | 210 (69) | 0.7344 |
| Yes | 118 (30) | 15 (29) | 103 (30) | | 25 (28) | 93 (31) | |
| **Self-medicated for diagnosed conditions, n (%)** | 61 (16) | 10 (20) | 51 (15) | 0.517 | 25 (28) | 36 (12) | **< 0.001** |
| **Awareness of black market, n (%)** | 94 (24) | 13 (25) | 81 (24) | 0.933 | 26 (29) | 68 (22) | 0.247 |
| **Geographical setting, n (%)** | | | | | | | |

(*Continued*)

**Table 1.** (Continued)

| Characteristics | Total N = 392 | Ever diverted prescription drugs, N = 51 | Did not divert prescription drugs, N = 341 | p value | Ever misused prescription drugs, N = 89 | Did not misuse prescription drugs, N = 303 | p value |
|---|---|---|---|---|---|---|---|
| Rural | 25 (6) | 4 (8) | 21 (6) | 0.132 | 8 (9) | 17 (6) | 0.218 |
| Urban | 226 (58) | 35 (69) | 191 (56) | | 55 (62) | 171 (56) | |
| Semi-urban | 141 (36) | 12 (24) | 129 (38) | | 26 (29) | 115 (38) | |

[a] n = 2: Missing data,

[b] n = 1: Missing data,

[c] n = 1: Missing data,

[d] n = 2: Missing data,

[e] n = 3: Not sure,

[f] n = 12: Missing data.

being homeless, tobacco use, illegal drug use and being aware of the black market, had no association with PM misuse (S3 Table).

## Association between ART adherence with prescription drug diversion and misuse

The reported ART adherence levels were greater than 95%, which is required to obtain a successful ART outcome. This was, however, not statistically significant as participants who were diverting PM had lower ART adherence (96%) when compared with participants who never diverted PM (97%) in the past seven days. Similarly, participants who were misusing PM had lower ART adherence (96%) when compared with participants who never misused PM (97%) in the past seven days. No regression analyses were performed since the ART adherence of all groups were relatively similar.

## Discussion

The researchers investigated the prevalence and correlates of prescription drug diversion, and misuse in a cohort of HIV-infected adults in the eThekwini district, KwaZulu-Natal province. The prevalence of prescription drug diversion among PLWH was 13% during their lifetime. Twenty three per cent of participants reported prescription drug misuse in the past 90 days.

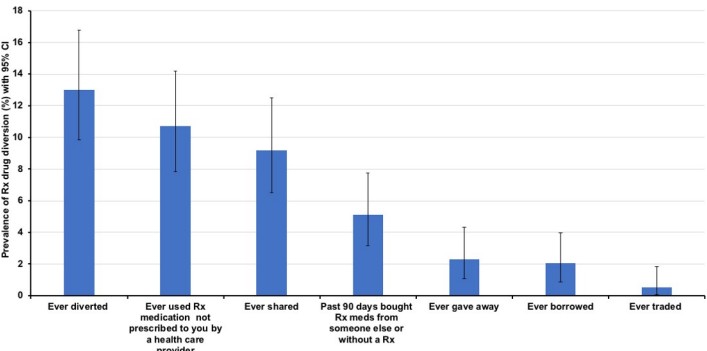

**Fig 1. The prevalence of diversion and types of prescription drug diversion, with a 95% confidence interval reported by people living with HIV in the eThekwini district.**

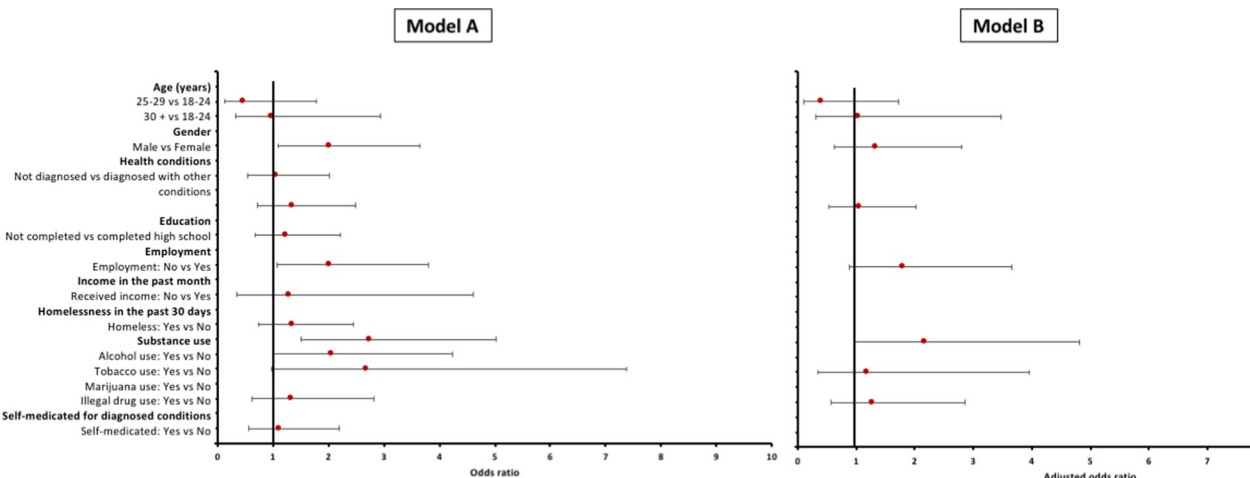

**Fig 2. Odds ratios showing the association of each variable on diversion among PLWH in the eThekwini district.** Model A represent the odds ratios showing the association of each variable on diversion among PLWH in the eThekwini district. All variables were fitted in the model, including geographical setting, to assess their collective association on the outcome for Model B. Model B represents adjusted odds ratios showing the association of a group of variables on diversion among PLWH in the eThekwini district.

Self-medicating was the only statistically significant predictor for prescription drug misuse among PLWH.

Although there is limited evidence on prescription drug diversion in resource-limited settings, numerous studies have been conducted in the United States where higher rates of prescription drug diversion have been reported among PLWH when compared to our findings [9]. A clinical trial conducted in the United States among men who have sex with men living with HIV on treatment documented that 27% of the participants reported to have ever sold their medication and 19% had diverted their treatment in the past year [17]. Prescription drug misuse is a silently growing problem globally, and literature has documented a wide prevalence spectrum, but our prevalence of prescription drug misuse is within the reported spectrum. Two longitudinal studies conducted in the United States among PLWH reported the prevalence of 21% opioid misuse in the past 90 days [42], and 37% abnormal use of prescription opioids in the past 90 days [43]. Other studies in the United States, however, reported lower prevalence when compared to our study: prevalence of 19% with a history of prescription drug misuse was reported among HIV-infected individuals in the US military [44] and a cross-sectional study in the United States among PLWH reported a prevalence of 11% of non-medical use of prescription opioids in the past year [45].

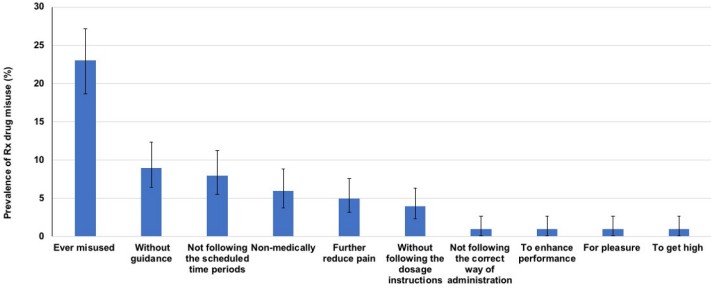

**Fig 3. The prevalence of misuse and the types of prescription drug misuse with a 95% confidence interval reported by people living with HIV in the eThekwini district.**

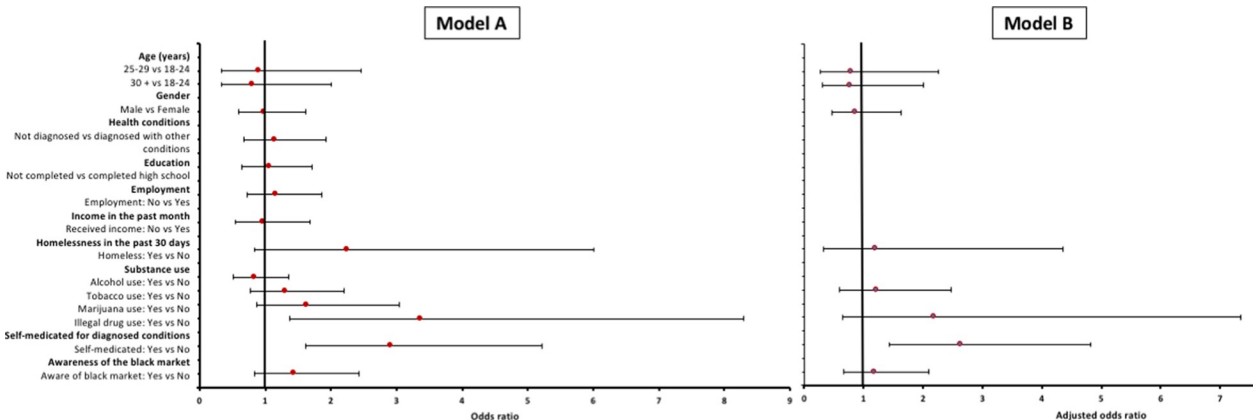

**Fig 4. Odds ratios showing the effect of each variable on misuse among PLWH in the eThekwini district.** Model A represent the odds ratios showing the association of each variable on misuse among PLWH in the eThekwini district. All variables were fitted in the model, including geographical setting, to assess their collective association on the outcome for Model B. Model B represents adjusted odds ratios showing the association of a group of variables on misuse among PLWH in the eThekwini district.

The prevalence of prescription drug diversion and misuse among PLWH is higher when compared with the general population. In support of this, according to the 2016 South African Demographic and Health Survey about 14% (women) and 15% (men) aged 15 years and older reported the use of codeine-containing medicines in the previous year however among these group, 2% of the women and 2% of the men reported non-medical use of codeine-containing products [46]. A recent study in South Africa reported 6% non-medical use of codeine-containing mixtures among the general population [30]. Results from the 2013 National Survey on Drug Use and Health documented 7% prescription drug misuse among adults [47] while current trend analysis reports a lower prevalence in 2018 when compared with rates reported in 2015–2017 among people aged 12 years or older [48]. Furthermore, in South Africa, codeine misuse was raised as a public health issue by health care providers since codeine-containing products were readily available in retail pharmacies and through the internet [31,32]. On the other hand, drug misusers in South African reported having exerted manipulative behaviour to influence healthcare workers in prescribing codeine-containing products [49].

Although these studies were conducted in upper-middle-income countries, an example in Jordan [50–52], China [53,54], Iran [55] and Turkey [56] among the general population instead of on PLWH, similar patterns of prescription drug misuse were observed, i.e. individuals stopped taking medications earlier than instructed [50,53], accessing prescription medicines without a doctor's medical script [51–55], using the medication without a medical indication [56], and taking higher dosages in the event of missed doses [50]. Increases in prescription drug overdose deaths have coincided with an increased rate of drug diversion, drug misuse and prescription drug-related illnesses [57–59]. Despite that, there was no association between ART adherence and prescription drug diversion and misuse in the past seven days in our study. Earlier research in the United States found that prescription opioid misuse was linked with less than 90% ART adherence [60], while a study conducted in 2017 in the United States reported that opioid use was significantly associated with a failure to achieve viral load suppression [61].

In South Africa, the Fifth South African National HIV Prevalence, Incidence, Behaviour and Communication Survey, (SABSSM V) of 2017, reported the highest HIV prevalence at 27% (95% CI 23.9–30.4) in KwaZulu-Natal province among people aged 15–49 years [62]. This shows that the population of PLWH is growing disproportionally at the provincial level. Our findings predict their vulnerability to prescription drug diversion, and misuse. This reflect a need for tailor-made interventions to protect PLWH when they seek health care.

Though self-medicating is a commonly practised behaviour, inappropriate self-medication practices have potential risks, not limited to the incorrect manner of drug administration, incorrect dosage consumption, incorrect choice of medicines, adverse drug reactions, risky drug-drug interactions, and risk of developing drug dependence and abuse [63]. This strongly supports the association between misuse and self-medicating. Inappropriate self-medicating might hinder progress to achieving and maintaining viral suppression outcomes among PLWH.

Our participants showed no engagement with selling their prescribed medications, but prescription drug diversion was commonly reported. Although the reason is unclear in our local context, men have been reported to delay seeking healthcare professional for any illness due to competing forces such as masculinity and stigma, however during the course of the illness men do obtain healing through other avenues which include getting medications from their partners, family, friends, traditional healers or from the black market place. Individuals aged 30 years and older had experienced challenges, barriers, stigma and discrimination during the early years of the HIV epidemic when there were conflicting information about the HIV epidemic and also the lack of political support which facilitated more stigma, slow development of HIV programmes, implementation of HIV management programmes and support programmes for affected persons. This has played a critical role in accessibility and utilization of healthcare systems, especially for males, due to inadequate or lack of male-tailored services.

In our local context, the household income does not play a major role in health care since most of our participants accessed health care from the public health sector where all services were free. Furthermore, substance use and dependency are known barriers to ART adherence since it disrupts healthcare engagement and retention in care, which later translates into incomplete HIV care. The implication of incomplete HIV care is unsuppressed viral loads; therefore, greater risk of transmitting the HIV virus resulting in increased HIV incidence rates and possibly AIDS-related deaths, due to opportunities infections. Due to the diverse dynamics of people living with HIV, one size fits all interventions to protect vulnerable individuals from prescription drug diversion, and misuse would not work.

The researchers recommend additional awareness interventions to supplement the medicine information pamphlet that is read and comprehended by only a few consumers. PLWH often present with known risk factors associated with prescription drug diversion, and misuse which include, but are not limited to, substance use, mental health problems and chronic pain. They frequently get prescribed controlled medications to alleviate experienced health problems. Therefore consideration of the correlates of prescription drug diversion, and misuse among PLWH is critical.

## Strengths and limitations

This study is among the first to investigate prescription drug diversion, and misuse among PLWH in the eThekwini district. Although the study consisted of sensitive questions, it provided valuable information on prescription drug diversion, and misuse. Behaviour practice-related questions were based on self-reports, and these may be impacted by recall and social desirability bias. The exclusion of a mental health variable in the analysis could be a limitation since mental health problems are one on the risk factors. Gaps were identified in the study questionnaire, and the questionnaire did not adequately cater for resource-limited settings, like the eThekwini district, since most people had low knowledge and awareness of different types of prescription drugs.

Due to limited human resource and funding to the study survey was only available in English, although these were the interviewer-administered questionnaires which allowed the researcher to assist participants who needed clarity in their preferred language. To this study's

advantage, the researcher was able to speak and understand most South African languages and all-dominating languages spoken in the KwaZulu-Natal province. The minimum sample size of 385 was determined using the formula developed by Daniel (1983) for a finite population with an assumption of 50% prevalence [64]. Due to limited funding for data collection, the study was able to enrol 392 PLWH, the small sample size to be representative PLWH in eThekwini district and attending healthcare services. A random sampling of public health clinics was employed, but there were inherent shortcomings due to convenient sampling. This approach is highly vulnerable to selection bias and influences beyond the control of the researcher has a great level of sampling error, and reliability can be compromised. Since some clinics refused to give permission to recruit participants at their facilities, the missed individuals might have different characteristics. The low variability in adherence scores (96% is impossible to really look at adherence in sub-analyses.)is a limitation, and future research should perhaps select PLWH with more diverse adherence profile. The type of prescription drugs diverted, and misused was excluded in the analysis since we only had data for the type of prescription drugs diverted. This posed as a limitation to our study analysis to compare the types of prescription drugs diverted, and misused. Comparisons were limited since the prevalence of prescription drug diversion, and misuse were investigated in different time periods in this study than in literature. Laboratory HIV viral load test results could be used to support the self-reported ART adherence. These findings may or may not be generalised beyond the eThekwini district among PLWH from resource-limited settings accessing care in public healthcare facilities, whereas prescription drug diversion, and misuse among PLWH accessing care in private facilities might be different.

## Conclusion

The high prevalence of prescription drug diversion and misuse among PLWH underscores the need for integrated surveillance tools at all levels in the healthcare system. Monitoring the dispensing of medicine and the evaluation of medicine usage by healthcare providers is highly recommended for routine prescriptions and for patients presenting with risk factors. The researchers recommend that education and awareness campaigns should be investigated. These campaigns need to be directed at patients or public knowledge of the potential risks and promoting greater awareness among healthcare providers and to the public. Caution should be taken when prescribing medicines with potential risks for diversion, and misuse especially when the patient is presenting with risk factors. This study estimates the prevalence of prescription drug diversion, and misuse among PLWH in the eThekwini district. Prescription drug diversion, and misuse practices need urgent attention since there is a risk of it transitioning into illegal drug use.

## Supporting information

**S1 Table. Characteristics participants for the study.**
(DOCX)

**S2 Table. Correlates of prescription drug diversion adjusted for geographical setting among PLWH in eThekwini district, participants who ever diverted prescription drugs, N = 51.**
(DOCX)

**S3 Table. Correlates of prescription drug misuse adjusted for geographical setting among PLWH in eThekwini district, participants who ever misused prescription drugs, N = 89.**
(DOCX)

**S1 Questionnaire. Prescription drug diversion, misuse and abuse among people living with HIV in eThekwini District, KwaZulu-Natal.**
(PDF)

**S1 Dataset.**
(CSV)

## Acknowledgments

The authors wish to acknowledge the participation of the following institutions/organisations: Human Sciences Research Council, KwaZulu-Natal Department of Health, TB HIV Care Organisation, as well all study participants and research assistants.

## Author Contributions

**Conceptualization:** Buyisile Chibi, Tivani P. Mashamba-Thompson.

**Data curation:** Buyisile Chibi.

**Formal analysis:** Buyisile Chibi, Nonhlanhla Yende-Zuma, Tivani P. Mashamba-Thompson.

**Funding acquisition:** Buyisile Chibi.

**Investigation:** Buyisile Chibi, Tivani P. Mashamba-Thompson.

**Methodology:** Buyisile Chibi, Tivani P. Mashamba-Thompson.

**Project administration:** Buyisile Chibi, Tivani P. Mashamba-Thompson.

**Resources:** Buyisile Chibi, Tivani P. Mashamba-Thompson.

**Supervision:** Buyisile Chibi, Tivani P. Mashamba-Thompson.

**Validation:** Buyisile Chibi, Nonhlanhla Yende-Zuma, Tivani P. Mashamba-Thompson.

**Visualization:** Buyisile Chibi.

**Writing – original draft:** Buyisile Chibi.

**Writing – review & editing:** Buyisile Chibi, Nonhlanhla Yende-Zuma, Tivani P. Mashamba-Thompson.

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
