## [Decision Letter · Decision Letter 0]

10 Feb 2020

PONE-D-19-31426

Prevalence and Predictors of Prescription Drug Diversion, Misuse and Abuse among People Living with HIV in eThekwini District, KwaZulu-Natal, South Africa

PLOS ONE

Dear Ms. Buyisile,

Thank you for submitting your manuscript to PLOS ONE. After careful consideration, we feel that it has merit but does not fully meet PLOS ONE’s publication criteria as it currently stands. Therefore, we invite you to submit a revised version of the manuscript that addresses the points raised during the review process.

Overall, reviewers thought this was an important topic and study (“really important and shows another angle in abuse and misuse of prescription medication”) but requires major revisions.  These reviews suggest two issues that should be the primary focus of the revision: 

Contextualization in the literature, especially from South AfricaDoing this will also requiring clarifying key points raised by the reviewersMake sure the terminology describing substance use is accurate and up to date for this field.More clear and accurate presentation of the data, and in some places consider re-running statistics (see comments on logistic regression model)Please consider carefully the comments on presentations of data and interpretation.Mare sure that the narrative carefully reflects the resultsPlease address sample size considerations

Generally, there are many small errors that need to be corrected.

There is a growing literature that examines the multiple challenges faced by people living with HIV which can meaningfully be used to inform policy and intervention, so I encourage the authors to use these thoughtful and specific reviews to improve this paper.

We would appreciate receiving your revised manuscript by Wednesday, March 25 2020. To enhance the reproducibility of your results, we recommend that if applicable you deposit your laboratory protocols in protocols.io, where a protocol can be assigned its own identifier (DOI) such that it can be cited independently in the future. For instructions see: http://journals.plos.org/plosone/s/submission-guidelines#loc-laboratory-protocols

We look forward to receiving your revised manuscript.

Kind regards,

Jennifer Zelnick

Academic Editor

PLOS ONE

Journal Requirements:

2. Please refer to any post-hoc corrections to correct for multiple comparisons during your statistical analyses. If these were not performed please justify the reasons. Please refer to our statistical reporting guidelines for assistance (https://journals.plos.org/plosone/s/submission-guidelines.#loc-statistical-reporting). In addition, please include additional information regarding the survey or questionnaire used in the study and ensure that you have provided sufficient details that others could replicate the analyses. For instance, if you developed a questionnaire as part of this study and it is not under a copyright more restrictive than CC-BY, please include a copy, in both the original language and English, as Supporting Information. This should also include further details of the pilot testing of the questionnaire, i.e. how many participants were involved and where they were recruited from.

3. We note that Figure 1 in your submission contain map images which may be copyrighted. All PLOS content is published under the Creative Commons Attribution License (CC BY 4.0), which means that the manuscript, images, and Supporting Information files will be freely available online, and any third party is permitted to access, download, copy, distribute, and use these materials in any way, even commercially, with proper attribution. For these reasons, we cannot publish previously copyrighted maps or satellite images created using proprietary data, such as Google software (Google Maps, Street View, and Earth). For more information, see our copyright guidelines: http://journals.plos.org/plosone/s/licenses-and-copyright.

You may seek permission from the original copyright holder of Figure 1 to publish the content specifically under the CC BY 4.0 license. 

If you are unable to obtain permission from the original copyright holder to publish these figures under the CC BY 4.0 license or if the copyright holder’s requirements are incompatible with the CC BY 4.0 license, please either i) remove the figure or ii) supply a replacement figure that complies with the CC BY 4.0 license. Please check copyright information on all replacement figures and update the figure caption with source information. If applicable, please specify in the figure caption text when a figure is similar but not identical to the original image and is therefore for illustrative purposes only.

6. Please upload a copy of Supporting Information Table 1 which you refer to in your text on page 7.

Reviewers' comments:

Reviewer's Responses to Questions

**Comments to the Author**

1. Is the manuscript technically sound, and do the data support the conclusions?

Reviewer #1: No

Reviewer #2: Yes

Reviewer #3: Partly

Reviewer #4: Partly

2. Has the statistical analysis been performed appropriately and rigorously? 

Reviewer #1: No

Reviewer #2: Yes

Reviewer #3: Yes

Reviewer #4: No

3. Have the authors made all data underlying the findings in their manuscript fully available?

Reviewer #1: Yes

Reviewer #2: Yes

Reviewer #3: Yes

Reviewer #4: Yes

4. Is the manuscript presented in an intelligible fashion and written in standard English?

Reviewer #1: Yes

Reviewer #2: Yes

Reviewer #3: Yes

Reviewer #4: No

5. Review Comments to the Author

Reviewer #1: Data presented in the results section does not match the narrative (comments attached). It is also not clear if the authors are discussing diversion and misuse of ARVs, or opioids used for pain management.

Reviewer #2: I think the study is really important and shows another angle in abuse and misuse of prescription medication; but I have some concerns about it:

*I dont believe that most of the research on this topic is mostly conducted in the USA only

*Reference provided about studies done in other countries (China, Jordan) show findings of prescription drug misuse in the general population and not im PWLH; therefore not similar to this study's findings

*There are several studies conducted in SA that looked at use and abuse of prescription medications; but these studies are not mentioned; and the statement mentioned about the study on misuse of codeine containig medicines use is not correct

*Description of 'misuse and abuse' is not clearly stated and that is confusing

*I appreciate the acknowledgement of excluding mental health variable as a limitation; i think the authors should have looked at it as is almost impossible to talk about HIV and exclude mental health (depression, anxiety, etc)

*Looks like figue 2 and figure 3 is the same graph and the y-axis heading is the same, confusing

*There are spelling mistakes, so the authors need to re-read the manuscript to avoid this

*I would have liked to see the names of commonly misused/diverted/abused medications; to see if these medications are mostly associated with mental health illnesses (to self-medicate)

Data analysis was conducted rigourously and that is appreciated. So i think the authors should be given another chance to look at corrections and resubmit the manuscript

Reviewer #3: The reviewer feels that it would be extremely valuable for the authors to engage further with literature around prescription and over the counter medication in low and middle income countries, but in South Africa particularly to get a sense of what some of the concerns and issues are in this context. Examples are publications on codeine data (authors to look for are Parry, C., Dada, S. and Carney, T.) and the SACENDU data which is widely available. There are also definitely at least a few studies on opioid use and HIV, which is increasingly a problem in KZN. Definitions of misuse and abuse and diversion are also not clearly provided. Please look at the latest terminology used in substance use literature.

It would have been very interesting to look at the relationship between HIV outcomes and problem prescription use among PLWH-could these types of analyses be run? There are some studies conducted in South Africa (one qualitative one actually conducted in KZN) that mention people may be misusing ART as well. Was this included in the current study?'

The results section could be presented in a more succinct way and it is difficult to interpret Figure 3 and 4. A suggestion would be to present the univariate and multiple regression models in one table-perhaps look at other literature for examples.

A study limitation is that this is a small sample size to be representative of eThekweni PLWH and attending healthcare services. Are there any additional comments to make around this, such as do the authors think that this is a representative sample?

Reviewer #4: The authors present results of a cross-sectional study on the prevalence and predictors of prescription drug diversion, misuse and abuse among individuals with HIV in a district within South Africa. The authors find that there is more misuse than drug diversion among their sample, and very little prescription drug abuse. They also identify a few risk factors for drug diversion and misuse. The manuscript will be strengthened if the authors consider the following points.

1. Figure 2: it is unclear why the confidence intervals presented in Figure 2 are not symmetric, since typical confidence intervals for proportions are symmetric. Authors should clarify how confidence intervals were computed and verify that what is presented in Figure 2 is correct.

2. lines 212-216 (also in the Abstract lines 36-40) and lines 248-250: The authors are reporting percentages out of all 392 participants, which isn't necessarily wrong, but the start of the sentences ("Majority" or "More than one-third") suggest that the reader should be focused on just those with drug diversion or with drug misuse. The authors are encouraged to think about what point they really want to make and whether it is most appropriate to report percentages out of the entire sample or out of just those that had the action of interest.

3. The authors have a tendency to overstate the findings, even when qualifying a statement by saying something wasn't statistically significant. In most of the situations, findings are not even close to being significant. Examples include lines 233-239, 241-244, 263-266, 272-275). Authors already include the ORs and 95% confidence intervals for the terms in the model in supplemental table 3, so readers can see these results. Some of these end up being discussed in the discussion (age and aware of the black market) and highlighted in the Abstract (homeless and awareness of the black market) as findings, which is inappropriate. In the discussion, they further highlight differences between males and females, but this finding does not hold up in the adjusted model.

4. In the adjusted models for drug diversion (model B), the authors include more variables than are generally recommended given the number of people who had the action of interest.Standard rules of thumb for logistic regression would suggest that no more than 5 predictors should be included in a model, yet the authors include 7.

Minor points:

1. line 27: "as mostly" should be "as most"

2. line 42: "diversion and misuse" should be "diversion or misuses" since not all factors were associated with both outcomes.

3. line 62: "South Africa account" should be "South Africa accounts" and consider changing "owning the" to "having the"

4. line 63: "has been" should be "have been"

5. line 70: "that correlates" should be "that correlate"

6. line 71: "suggest" should be "suggests"

7. line 73: "and abuse has been" should be "and abuse have been"

8. line 83: what do the authors mean by saying trends are close to non-existence (or non-existent) - do they mean there is little information?

9. lines 86-87: the sentence starting with "Furthermore" is an incomplete sentence

10. line 106: remove "until"

11: line 130: "Data was" should be "Data were"

12. line 144: "development the" should be "development of the"

13.lines 145-146: the sentence "The emphasises the importance..." is unclear and authors are encouraged to reword it for clarity.

14. line 148: "diversion stem" should be "diversion which stems"

15. line 150: "is prescription drug misuse was" should be " is prescription drug misuse and was"

16. line 151: "drug misuse stem" should be "drug misuse stems"

17. line 185: "rigour strategy" should be "rigorous strategy"

18. line 193: "Analysis were" should be "Analyses were"

19: line 203: the authors report a response rate of 102%, but it is unclear how a response rate can be larger than 100%.

20. Figure 2 and 4 appear to be identical, so authors should verify that the correct figure was uploaded for Figure 4.

21. lines 287-291: these lines should be reworded given that the percentages are so similar.

22. lines 298-299 - authors should be clear this result is from the adjusted model

23. line 328: "with an increase rates" should be "with increased rates"

24: line 349: "This strongly support" should be "This strongly supports"

25. line 361: "metal" should be "mental"

6. PLOS authors have the option to publish the peer review history of their article (what does this mean?). If published, this will include your full peer review and any attached files.

Reviewer #1: No

Reviewer #2: No

Reviewer #3: No

Reviewer #4: No

---

## [Author Response · Author response to Decision Letter 0]

9 Apr 2020

Rigorous editing was done to the entire paper.

Definition section was added.

The narrative of the results section was revised to match the data.

The type of prescription drugs diverted, misused and abused were excluded in the analysis since we only had data for type of prescription drugs diverted and abused. This posed as a limitation to our study analysis to list and compare the types of prescription drugs diverted, misused or abused.

The authors had taken note that the p-values were >0.05 (no association). However, our participants with the following characteristics (homelessness, used tobacco, used marijuana and are of the black had higher chances of misusing prescription medication. 

Studies conducted in South Africa were included.

The study was self-report and most participants did not know their latest viral load. We were unable to evaluate the HIV outcomes.

Regression models A and B were presented in one table for prescription drug diversion and the same was done for prescription drug misuse. 

Mode B for prescription drug diversion and misuse with only five variables was redone. However, we found that there was no significant difference between the original model and revised model looking at the Odds ratios. Therefore both Model B figures were unchanged. 

Due to limited funding for data collection the study was only able to enroll 392 PLWH. However The minimum sample size of 385 was determined using the formula developed by the Daniel (1982) for a finite population with an assumption of 50% prevalence.

---

## [Decision Letter · Decision Letter 1]

9 Jul 2020

PONE-D-19-31426R1

Prevalence and Predictors of Prescription Drug Diversion, Misuse and Abuse among People Living with HIV in eThekwini District, KwaZulu-Natal, South Africa

PLOS ONE

Dear Dr. Chibi,

Thank you for submitting your manuscript to PLOS ONE. After careful consideration, we feel that it has merit but does not fully meet PLOS ONE’s publication criteria as it currently stands. Therefore, we invite you to submit a revised version of the manuscript that addresses the points raised during the review process.

We look forward to receiving your revised manuscript.

Kind regards,

Becky L. Genberg

Academic Editor

PLOS ONE

Additional Editor Comments (if provided):

While some of the concerns from the previous review have been addressed, there remain concerns by one reviewer.

Additionally, I have concerns regarding the definitions of prescription drug abuse, self-medication, and diversion. Firstly, what drugs specifically are you inquiring about? the potential for misuse or abuse and the implications for HIV prevention and treatment would apply only to specific classes of prescription drugs. Were these assessed? Secondly, the definition of abuse is not in line with the substance use literature and should be reconsidered. Please clarify that 100% response rate means that all persons approached to participate agreed and completed the survey. Finally I also share concerns of the reviewer about the need for careful editing, as some numbers appear incorrect and there are several typographical and grammatical errors throughout the manuscript that remain in this revision. In this revision I would recommend adhering to commonly accepted terms when referring to people who use drugs.

Reviewers' comments:

Reviewer's Responses to Questions

**Comments to the Author**

1. If the authors have adequately addressed your comments raised in a previous round of review and you feel that this manuscript is now acceptable for publication, you may indicate that here to bypass the “Comments to the Author” section, enter your conflict of interest statement in the “Confidential to Editor” section, and submit your "Accept" recommendation.

Reviewer #2: All comments have been addressed

Reviewer #4: (No Response)

2. Is the manuscript technically sound, and do the data support the conclusions?

Reviewer #2: Yes

Reviewer #4: Yes

3. Has the statistical analysis been performed appropriately and rigorously? 

Reviewer #2: Yes

Reviewer #4: Yes

4. Have the authors made all data underlying the findings in their manuscript fully available?

Reviewer #2: Yes

Reviewer #4: Yes

5. Is the manuscript presented in an intelligible fashion and written in standard English?

Reviewer #2: Yes

Reviewer #4: Yes

6. Review Comments to the Author

Reviewer #2: (No Response)

Reviewer #4: The authors have addressed the majority of my earlier concerns. There are a few remaining issues that should be addressed.

1. line 103: "might unfolds" should be "might unfold"

2. lines 122-123: "areas with lack training" should possibly be "areas which lack tracking"

3. line 207: "would their participation affect their healthcare" should be "and participation would not affect their healthcare"

4. line 280: mention these are exact 95% confidence intervals

5. Table 1: authors should carefully check numbers and p-values as I found a number of incorrect calculations. Examples include the percentage for 18-24 years in the Total sample (which should be 10% not 7%), the percentage of completed high school and tertiary studies in the did not misuse prescription drugs group (which should be 10% not 26%), the numbers of homeless add to 18, not 17 when looking at the yes/no columns for both divert and misuse prescriptions, the p-value for ever used alcohol comparing those that diverted and those that did not seems wrong (based on the numbers provided), I get a much smaller p-value for ever used tobacco in those that diverted vs not, the numbers of ever used marijuana in the diverted and did not divert columns add to 56 not 55.

6. It is recommended that in the text (lines 339, 360, 390) and S2 and S3 tables, the authors remind the readers what model A and model B mean. At a minimum, there should be a note for S2 and S3 tables that defines these models.

7. lines 350-356 are all results that have extremely high p-values, so reporting a 20% or 50% increase in odds is misleading. There is no evidence in the data to suggest that there is any increased chance of diverting PM based on these characteristics. I am fine with keeping the marijuana and illegal drugs findings, since with the 95% confidence interval, the readers can see that the lower limit for the CI is just under 1.

8. lines 365 and 367, remove "among the participants who misused drugs", because the authors are reporting percentages out of the entire sample not just those who misused drugs.

9. lines 321-385: it is misleading to say that these have higher odds of misusing PM, since the results all have p-values >0.1.

7. PLOS authors have the option to publish the peer review history of their article (what does this mean?). If published, this will include your full peer review and any attached files.

Reviewer #2: No

Reviewer #4: No

---

## [Author Response · Author response to Decision Letter 1]

25 Aug 2020

All reviewers suggestions were considered.

All analysis were redone to double check any errors. Table 1, supplementary information and Fig 1-4 were update.

Definitions were updated according to the National Institute on Drug Abuse.

Prescription drugs inquired about in the study were sedatives, stimulants, analgesics, antibiotics, antiretroviral drugs and any emerging prescription drugs with potential of being diverted, misused or abused. 

For example:

Any sedatives (tranquilizers) such as Xanax, Valium, Ativan, Klonopin, Soma, Barbiturates or others 

• For euphoric effects such as excitement and happiness 

Any stimulants such as Adderall, Ritalin, Adipex or any other stimulants 

• For increasing the ability to concentrate, boost feelings and performance enhancing 

Any analgesics (painkillers) such as Opioids, Codeine, Tramadol, percocet, Vicodin, OxyContin, , Demerol, Darvon, Demerol, Morphine, Methadone, Fentanyl, Pentazocine

• For reducing pain or feeling high 

Any antibiotics such as amoxicillin, ampicillin, cloxacillin, metronidazole, co- trimoxazole, ciprofloxacin, or others 

Any ARVS

Were the prescription drugs assessed?

There was a limitation in the study since most participants had low knowledge and awareness of different types of prescription drugs. Participants were unable to specify the exact name of the prescription medication.

Grammarly was used to improve the manuscript.

---

## [Editor Report · Decision Letter 2]

16 Sep 2020

PONE-D-19-31426R2

Prevalence and Predictors of Prescription Drug Diversion, Misuse and Abuse among People Living with HIV in eThekwini District, KwaZulu-Natal, South Africa

PLOS ONE

Dear Dr. Chibi,

Thank you for submitting your manuscript to PLOS ONE. After careful consideration, we feel that it has merit but does not fully meet PLOS ONE’s publication criteria as it currently stands. Therefore, we invite you to submit a revised version of the manuscript that addresses the points raised during the review process.

ACADEMIC EDITOR:

 Thank you for addressing most of the concerns from the previous review. There are some remaining concerns:

    1) The legends (or footnotes) for Figures 2 and 4 should specify the difference between Model A and Model B.

    2) In the analysis section, the statements about the selection of factors included in the model should be appropriately referenced. For example, where in lines 329-30 commenting on "known confounders" of the exposure-outcome relationship. Also, with respect to the language in this paragraph; these are associations, not effects, between the variables and the outcomes. These factors are also more appropriately termed correlates, not predictors.

    4) The assessment of "drug abuse" is not an accepted method of measurement, and the terminology is outdated. The idea that one could assess this type of use by directly asking about use to intentionally harm oneself lacks the understanding that addiction is a behavioral pattern in which one uses substances despite the negative consequences. I highly recommend removing this from the paper and focusing on diversion and misuse and the types of misuse examined.

We look forward to receiving your revised manuscript.

Kind regards,

Becky L. Genberg

Academic Editor

PLOS ONE

---

## [Author Response · Author response to Decision Letter 2]

26 Sep 2020

Reviewer comment 1: The legends (or footnotes) for Figures 2 and 4 should specify the difference between Model A and Model B.

Author response 1: Description of model A and B were explained in the legends for bot drug diversion and misuse.Page 25, line 382-387; Page 26, line 415-421

Reviewer comment 2: 2) In the analysis section, the statements about the selection of factors included in the model should be appropriately referenced. For example, where in lines 329-30 commenting on "known confounders" of the exposure-outcome relationship. Also, with respect to the language in this paragraph; these are associations, not effects, between the variables and the outcomes. These factors are also more appropriately termed correlates, not predictors.

Author response 2: Reference 40 and 41 were added; Page 14, line 318. The word predictors was replaced with correlates in the manuscript title; Page 1, line 1. The word effect was replaced with associations. Page 13, line 308-317

Reviewer comment 3: 4) The assessment of "drug abuse" is not an accepted method of measurement, and the terminology is outdated. The idea that one could assess this type of use by directly asking about use to intentionally harm oneself lacks the understanding that addiction is a behavioral pattern in which one uses substances despite the negative consequences. I highly recommend removing this from the paper and focusing on diversion and misuse and the types of misuse examined.

Author response 3: Suggestion taken into consideration. All information related to drug abuse was removed from the paper.

---

## [Editor Report · Decision Letter 3]

30 Nov 2020

Prevalence and correlates of prescription drug diversion and misuse among people living with HIV in the eThekwini district, KwaZulu-Natal, South Africa

PONE-D-19-31426R3

Dear Dr. Chibi,

We’re pleased to inform you that your manuscript has been judged scientifically suitable for publication and will be formally accepted for publication once it meets all outstanding technical requirements.

Kind regards,

Dawn K. Smith

Academic Editor

PLOS ONE

Additional Editor Comments (optional):

Thank you for the revisions in response to reviewer comments.

---

## [Editor Report · Acceptance letter]

7 Dec 2020

PONE-D-19-31426R3 

Prevalence and correlates of prescription drug diversion and misuse among people living with HIV in the eThekwini district, KwaZulu-Natal, South Africa 

Dear Dr. Chibi:

I'm pleased to inform you that your manuscript has been deemed suitable for publication in PLOS ONE. Congratulations! Your manuscript is now with our production department. 

Kind regards, 

on behalf of

Dr. Dawn K. Smith 

Academic Editor

PLOS ONE